# Development of an Earth-Field Nuclear Magnetic Resonance Spectrometer: Paving the Way for AI-Enhanced Low-Field Nuclear Magnetic Resonance Technology

**DOI:** 10.3390/s24175537

**Published:** 2024-08-27

**Authors:** Eduardo Viciana, Juan Antonio Martínez-Lao, Emilio López-Lao, Ignacio Fernández, Francisco Manuel Arrabal-Campos

**Affiliations:** 1Department of Engineering, Research Centre CIMEDES, Escuela Superior de Ingeniería, University of Almería, 04120 Almeria, Spain; evg010@ual.es (E.V.); ell658@inlumine.ual.es (E.L.-L.); 2Department of Chemistry and Physics, Research Centre CIAIMBITAL, University of Almería, 04120 Almeria, Spain

**Keywords:** Earth’s field NMR, NMR spectrometer, EFNMR

## Abstract

Today, it is difficult to have a high-field nuclear magnetic resonance (NMR) device due to the high cost of its acquisition and maintenance. These high-end machines require significant space and specialist personnel for operation and offer exceptional quality in the acquisition, processing, and other advanced functions associated with detected signals. However, alternative devices are low-field nuclear magnetic resonance devices. They benefit from the elimination of high-tech components that generate static magnetic fields and advanced instruments. Instead, they used magnetic fields induced by ordinary conductors. Another category of spectrometers uses the Earth’s magnetic field, which is simple and economical but limited in use. These devices are called Earth-Field Nuclear Magnetic Resonance (EFNMR) devices. This device is ideal for educational purposes, especially for engineers and those who study nuclear magnetic resonance, such as chemistry or other experimental sciences. Students can observe their internal workings and conduct experiments that complement their education without worrying about damaging equipment. This article provides a detailed explanation of the design and construction of electrical technology devices for the excitation of atomic spin resonance using Earth’s magnetic fields. It covers all necessary stages, from research to analysis, including simulation, assembly, construction of each component, and the development of comprehensive software for spectrometer control.

## 1. Introduction

The history of nuclear magnetic resonance (NMR) is deeply intertwined with advances in multiple scientific disciplines, leading to its application in various fields such as chemistry, biochemistry, and medicine [1]. The initial breakthroughs in NMR can be traced back to the quantization of nuclear spin angular momentum observed by Stern and Gerlach in 1936 and the magnetic resonance method developed by Rabi and his team, which allowed for the precise measurements of magnetic moments [2,3]. Despite early challenges, such as Gorter’s unsuccessful attempts to detect NMR in solids due to long relaxation times, these foundational experiments set the stage for further developments [1].

From the 1940s through the 1950s, significant strides in NMR technology were made. Bloch and Purcell, working independently, were the first to detect NMR signals in bulk matter, a discovery that earned them the Nobel Prize in Physics in 1952 [4,5]. Their work revealed much shorter relaxation times than previously anticipated, marking a pivotal moment in the understanding and application of NMR [6]. During this period, innovations such as the use of short radiofrequency pulses to excite nuclear spins and the discovery of spin echo by Hahn further refined NMR techniques, particularly in measuring the T_1_ and T_2_ relaxation times [7,8,9]. These advances were crucial in developing a new theoretical framework for NMR, including Bloch’s introduction of the relaxation times and the BPP theory, which explained the effects of molecular motion on NMR signals [10,11].

The application of NMR has expanded rapidly with the discovery of chemical shifts and scalar coupling. Early observations, such as the two distinct resonance signals in *^14^N* with $NH_{4}NO_{3}$ and the identification of chemical shifts in various compounds, laid the foundation for using NMR in the determination of chemical structures [12,13,14,15]. The introduction of the term “chemical shift” by Herbert S. Gutowsky and the correlation of shifts with structural features significantly impacted the utility of NMR in chemistry [16]. High-resolution NMR techniques revealed the multiplicity of chemically shifted signals, further enhancing their application in structural analysis [17,18].

Dynamic nuclear polarization (DNP), discovered by Overhauser in 1953, and its subsequent verification and application in the nuclear Overhauser effect (NOE) by Solomon, provided new tools to enhance signal intensity in NMR, although these were not widely recognized until later [19,20,21]. As NMR technology matured, commercial spectrometers, such as those developed by Varian Associates, played a crucial role in making NMR more accessible to the scientific community [22,23]. Advances such as the Fourier transform NMR (FT-NMR) by Anderson and Ernst in the 1960s revolutionized data acquisition, allowing for faster and more efficient analysis, which was recognized with a Nobel Prize in Chemistry in 1991 [24].

NMR technology continued to advance through the 1970s and 1980s, with the development of two-dimensional (2D) NMR, proposed by Jean Jeener and advanced by Richard Ernst [25,26]. These techniques allowed for more detailed structural analyses, particularly in the study of biopolymers and complex molecules [27,28,29]. The integration of computational methods and improvements in hardware and software further enhanced the capabilities of NMR, making it a powerful tool for studying dynamic processes and the structure of biological substances [30].

In the field of medicine, the application of NMR took a significant leap with the introduction of magnetic resonance imaging (MRI) in the 1970s. The work of Lauterbur on spatial encoding and Mansfield on echoplanar imaging (EPI) led to the development of non-invasive diagnostic tools that revolutionized clinical medicine [31,32]. Solid-state NMR (ssNMR) also advanced, enabling the study of complex solid materials such as biological membranes and polymers [33,34].

The first Earth’s field NMR (EFNMR) was performed in 1954 [35,36]; it was a thing of curiosity, and the main use was for geophysical magnetometry [37] or simply a low-cost demonstration of magnetic resonance principles. Despite its low field, it continues to attract the attention of researchers and engineers not only because of its rarity as a physical phenomenon but also because of certain low-field results that are difficult to obtain in a high field, such as the inflection obtained in the dependence of the inverse of T1 below 0.01 T of the signal detected in some tissues [38], as well as other applications encountered [39,40]. The strength of the Earth’s magnetic field is very low. It is still possible to obtain narrow NMR lines and a high signal-to-noise ratio with an acceptable spectral resolution. In the 1960s, Béné and colleagues demonstrated that EFNMR could measure the *J*-coupling of nuclear elements [41]. Because high homogeneity and stability can be achieved successfully in low fields, experiments have proven that many important spin multipliers can be studied in low fields. Chemical shift differences have recently been demonstrated to be measured even in inhomogeneous low magnetic fields [42].

This device presents several potential applications. One significant application is the precise measurement of the chemical shift of hyperpolarized xenon-129 in the Earth’s magnetic field, achieving a precision similar to high-field measurements [43]. Furthermore, ^1^H and ^19^F EFNMR spectra can be acquired with frequency resolutions nearly two orders of magnitude better than those obtainable with high-field high-resolution NMR [44]. Another application of the breakthrough was the observation of heteronuclear J-couplings using the classic COSY experiment [45], and more recently, the first observation of homonuclear proton J-couplings in the Earth’s magnetic field, previously considered impossible [46]. In the realm of condensed matter physics, low-field NMR allows for high-resolution spectra in solids by minimizing the line broadening caused by dipole–dipole interactions in high fields [47]. Moreover, low magnetic fields allow for the observation of “forbidden transitions”, providing unique information that is not detectable at high magnetic fields Kohl et al. [48]. These applications highlight the versatility and potential of NMR in various scientific fields.

## 2. Motivation and Objectives

NMR has proven to be an exceptionally valuable tool in a variety of disciplines, from diagnostic medicine to chemical and biological research. However, most current NMR spectrometers require intense magnetic fields, which limits their use to specialized laboratory environments. The development of an NMR spectrometer that can effectively operate using the Earth’s magnetic field represents a significant pedagogical shift. This advancement could revolutionize the educational application of NMR, enabling its integration into specialized NMR courses within the curriculum.

Current NMR equipment are prohibitively expensive for many institutions, especially those in developing countries and organizations with limited budgets. Using the Earth’s magnetic field, a more accessible NMR spectrometer can significantly reduce costs and lower barriers to access, allowing for a much broader range of researchers to use this valuable tool. However, its use is currently limited to educational purposes.

The potential applications of an Earth-field NMR spectrometer extend beyond traditional laboratories. There are numerous situations where access to conventional NMR equipment is challenging or impossible, such as field research, low-resource environments, or remote geographic regions. An Earth-field NMR spectrometer could enable on-site analyses, provide real-time data, and eliminate the need for sample transportation. As it can be transported, this has allowed us to carry out extensive research of diffusion in explorations of Antartic ice and groundwater [49,50].

Research into an Earth-field NMR spectrometer also has the potential to deepen our fundamental understanding of nuclear magnetic resonance. By working with much lower magnetic field intensities than usual, we can gain new insights into the underlying principles of NMR and uncover previously unobserved behaviors and properties. This additional knowledge could open new avenues for research and application beyond the purely educational purposes mentioned above.

The main objective of this paper is to design and construct an NMR spectrometer that uses the Earth’s magnetic field as a constant magnetic field to operate. In addition, the aim is to design systems that can generate acceptable and useful results under various operating conditions. To evaluate the efficiency and effectiveness of the Earth-field NMR spectrometer, its results are compared to those of conventional NMR spectrometers. One of the main advantages of Earth-based NMR spectrometers is the ability to use NMR spectrometers in situations where conventional NMR spectrometers cannot be used. Consequently, spectrometers are designed and developed so that they are as portable, durable, and user-friendly as possible to be used in a wider range of environments. Data collected by Earth’s field NMR spectrometers must be processed and analyzed. Herein, we developed robust and user-friendly software systems to manage this process, simplify data interpretation, and allow users to extract the maximum value from the results obtained.

## 3. Principle of Operation and General Aspects

Nuclear magnetic resonance (NMR) is a physical phenomenon in which atomic nuclei in a magnetic field absorb and re-emit electromagnetic radiation. This process allows for an investigation of the physical and chemical properties of the nucleus itself and its surrounding environment.

Nuclear spin is a fundamental property of atomic nuclei related to the number of protons and neutrons they contain. Some nuclei with specific properties, such as ^1^H (protons), ^13^C, ^15^N, and ^31^P, are NMR active, which means they have a nonzero spin quantum number. This spin imparts a magnetic moment to them, turning them into tiny magnetic bars.

In an NMR experiment, an external magnetic field, $B_{0}$, is applied. The magnetic moments of the nuclei align with the external field, but in quantum mechanics, this alignment is quantized; Figure 1. This means that nuclear spins can align in specific orientations dictated by their spin quantum number ($m_{l}$). For example, a nucleus with a spin quantum number of 1/2 (such as ^1^H or ^13^C) has two possible spin states in an external magnetic field: “spin-up” ($m_{l}=+1/2$) or “spin-down” ($m_{l}=-1/2$). These states have different energies, with the spin-down state being the higher energy state.

The constant that relates the Larmor frequency to the applied magnetic field is called the gyromagnetic constant, which, for *^1^H*, is 42.574 MHz/T. The energy levels produced can be quantified as follows:(1)$$E_{I}=h\nu_{I}=hm_{I}\gamma_{I}B_{0}$$
where $E_{I}$ is the energy level of the atom *I*, *h* is Planck’s constant, $m_{I}$ is the quantum spin number, $\gamma_{I}$ is the gyromagnetic constant, and $B_{0}$ is the external magnetic field. These two energy levels are known as the alpha and beta states. For the hydrogen atom, the energy levels ($E_{I}$) for a 600 MHz (14.1 T) spectrometer are around $2\times 10^{-25}$ J. The thermal energy ($E_{T}$) that determines the population ratio of a partition distribution is around $4.1\times 10^{-21}$ J. Thus, the population associated with this energy state can be calculated as follows:(2)$$N_{I}=\frac{1}{2}e^{\left(E_{I}/E_{T}\right)}=\frac{1}{2}e^{\left(\frac{m_{I}h\gamma_{I}B_{0}}{k_{b}T}\right)}$$

The population ratio for the different levels can be estimated by subtracting the energy levels and dividing them by the sum of the two:(3)$$\frac{N_{+1/2}-N_{-1/2}}{N_{+1/2}+N_{-1/2}}=\frac{e^{\frac{h\gamma_{I}B_{0}}{k_{b}T}}-e^{\frac{-h\gamma_{I}B_{0}}{k_{b}T}}}{e^{\frac{h\gamma_{I}B_{0}}{k_{b}T}}+e^{\frac{-h\gamma_{I}B_{0}}{k_{b}T}}}=\tanh\left(\frac{h\gamma B_{0}}{2k_{b}T}\right)\approx \frac{h\gamma B_{0}}{2k_{b}T}$$

Equation (Equation 3) reveals that for a 600 MHz spectrometer at 288 K, the fraction of populations is around $\frac{\Delta N_{\pm 1/2}}{N}\approx 5\times 10^{-5}$; therefore, only a small fraction will have a net magnetization $M_{0}$:(4)$$M_{0}=N\frac{h\gamma }{2}\left(\frac{h\gamma B_{0}}{2k_{b}T}\right)=\frac{C_{m}VB_{0}}{T}\left(\frac{h^{2}\gamma^{2}}{4R}\right)$$
where *N* is the total number of populations, $C_{m}$ is the molar concentration of the nuclei considered, *V* is the volume of the sample, $B_{0}$ is the external magnetic field, and *R* is the universal gas constant.

$M_{0}$ is the projection on the Z-axis parallel to the external magnetic field, referred to as the polarization in $M_{z}$ of all spins, which defines the signal intensity in resonance. These energy levels are called Zeeman energy levels and determine the magnetization parallel to the external magnetic field (Figure 2). In NMR, a radiofrequency (RF) pulse is used to perturb the nuclear spins from their equilibrium state. If this RF pulse matches the Larmor frequency (a condition known as “on resonance”), the nuclear spins can absorb this energy and “flip” from the lower-energy state to the higher one. Once the RF pulse is turned off, the system will return to equilibrium, a process known as relaxation. During this process, the nuclei will emit the absorbed energy in the form of an electromagnetic signal, which can be detected and used to generate the NMR spectrum (Figure 2). The population difference between the alpha and beta states is very small but sufficient to produce an observable NMR signal, as explained previously. The system is in equilibrium when the ratio between the populations of the alpha and beta states remains constant. In this state, there is no net magnetization in the transverse plane (x-y). A short and intense burst of radiofrequency (RF) radiation, called an RF pulse, is applied perpendicular to $B_{0}$. If this RF pulse matches the Larmor frequency (the resonance frequency of the spins), the nuclear spins can absorb this energy, causing a 90° flip of their alignment with $B_{0}$ into the transverse plane (x-y). This generates a net magnetization in the transverse plane, creating a non-equilibrium state (Figure 2). It is a natural process based on the minimization of energy and maximization of entropy; see Table 1. In NMR, this natural process is observed by minimizing the spin energy and maximizing the nuclear spin phase.

The variables involved in the spin perturbation process are the intensity of the radiofrequency pulse ($B_{RF}$), the phase value, and the duration of the pulse ($\tau_{p}$). The flip angle of the Z-axis over the X-Y plane is determined as $\phi_{p}$ = $\tau_{p}\gamma B_{RF}$.

After turning off the RF pulse, the system does not remain in a non-equilibrium state. The spins gradually return to their equilibrium alignment along the Z-axis, a process known as relaxation. Relaxation occurs in two distinct processes: longitudinal relaxation (or T_1_) and transverse relaxation (or T_2_).

T_1_ relaxation is an enthalpic process where a net electromagnetic emission occurs, producing a detectable signal as the system transitions from a non-equilibrium state to equilibrium. Also known as spin-lattice relaxation, T_1_ relaxation is the process in which the net magnetization vector in the transverse plane returns to the Z-axis. This process is associated with the energy released by the spins as they return to equilibrium, causing the nuclei to revert to their original energy states. The relaxation time T_1_ is a measure of how quickly this process occurs. During this relaxation process, the detectable signal mentioned earlier is observed as what is known as Free Induction Decay (FID). FID is the time-dependent signal generated by the precessing net magnetization vector in the transverse plane immediately after the initial radiofrequency (RF) pulse is applied. As the magnetization vector relaxes back to its equilibrium state along the Z-axis, the FID signal decays exponentially, capturing the transition of the system back to equilibrium. The physical model of relaxation can be mathematically expressed as follows:(5)$$M_{z}=M_{z0}\left(1-e^{-t/T_{1}}\right)$$
(6)$$M_{x}=M_{x0}\sin\left(t\nu_{H}\right)e^{-t/T_{1}}$$
(7)$$M_{y}=M_{y0}\cos\left(t\nu_{H}\right)e^{-t/T_{1}}$$

Equation (5) shows what is called a coherent transition where photons are emitted, resulting in a detectable signal. The relaxation T_2_ is an entropic process in which the detectable signal has a phase among the different nuclear spins, producing relaxation in the XY plane. Also known as spin–spin relaxation, T_2_ relaxation is the process by which the spins in the XY plane dephase (i.e., lose their coherence). During this process, the transverse magnetization decays to zero as a result of interactions between the spins. The relaxation time T_2_ is a measure of the rate at which the transverse magnetization decays. A complete visualization of this relaxation process can be found in the following references for different experiments [7,8,51,52].
(8)$$M_{z}=M_{z}\left(1-e^{-t/T_{1}}\right)$$
(9)$$M_{x}=M_{x}\sin\left(t\nu_{H}\right)e^{-t/T_{2}}$$
(10)$$M_{y}=M_{y}\cos\left(t\nu_{H}\right)e^{-t/T_{2}}$$

This is the classical description of nuclear magnetization behavior developed by Bloch in his vector model in the presence of an external magnetic field and following excitation by an RF pulse. These equations, named after Felix Bloch, model nuclear spins as a bulk magnetization vector and describe their precession and relaxation behaviors. Although Bloch’s model is incredibly useful and has greatly contributed to our understanding of NMR, it has several limitations.

Lack of quantum effects: Bloch equations are classical and do not account for quantum mechanics effects. For example, they do not consider that the spin state of a nucleus is quantized.Assumption of a homogeneous magnetic field: The equations assume a perfectly homogeneous external magnetic field. In reality, magnetic fields often exhibit inhomogeneities, which can lead to various effects not covered by Bloch equations.Assumption of instantaneous excitation: The equations assume that the transition from equilibrium to non-equilibrium (during the RF pulse) is instantaneous. In real scenarios, the RF pulse has a finite duration, and its effect on the spin system might not be as abrupt as assumed by Bloch equations.Neglect of multiple spin interactions: Bloch’s model does not consider the interaction of different spins with each other (spin–spin interactions), which can lead to more complex phenomena such as relaxation and spin–spin coupling.Assumption of large spin ensembles: Bloch equations consider the behavior of a spin ensemble, thus providing average values. They do not provide information on the behavior of individual spins within the ensemble.Simplification of relaxation processes: The equations simplify relaxation processes (T_1_ and T_2_) to single exponential decays, whereas in reality these processes can involve multiple components with different relaxation times.

Despite these limitations, the Bloch equations remain a fundamental tool in the understanding and interpretation of NMR, offering a simplified and efficient description of many phenomena encountered in this field. However, for a more detailed understanding of complex NMR phenomena, it is necessary to resort to quantum mechanical descriptions or more advanced models.

Nuclear magnetic resonance in the geomagnetic field is known as Earth’s Field NMR (EFNMR), a special case of low-field NMR. When a sample is placed in a uniform magnetic field, nuclei with non-zero spin experience resonance at specific frequencies. This document exclusively covers proton nuclear magnetic resonance (NMR), primarily involving the ^1^H isotope. Moreover, the experiments described in this study depend on the use of a polarizing coil to achieve the initial magnetization of the sample, which serves as an alternative to the traditional RF excitation pulse commonly used in high-field NMR. This choice is particularly important in the context of Earth-field NMR, where the magnetic field strength is extremely low, and consequently, the thermal polarization of the spins is also weak. The polarizing coil provides a stronger, more uniform magnetic field during the polarization phase, effectively increasing the initial magnetization of the sample before signal detection. This is very important in low-field conditions, where the natural magnetization would, otherwise, be insufficient to produce a detectable signal. This device uses separate coils for polarization and reception; each can be optimized independently for its specific function. The polarizing coil is designed to generate a strong and uniform magnetic field, while the receiving coil is optimized for maximum sensitivity and signal-to-noise ratio during detection. This approach enhances the overall performance of the NMR system. Utilizing a dedicated polarizing coil helps to avoid potential artifacts or residual signals from the excitation phase that could interfere with the detection phase. This results in cleaner and more accurate signal acquisition, which is particularly important for reliable measurements in an Earth-field NMR setup.

## 4. Materials and Methods

This document exclusively covers proton NMR, mainly involving the ^1^H isotope. In the Earth’s magnetic field, which is around 45,000 nT depending on the location, the ^1^H isotope has a Larmor frequency of approximately 2 kHz and generates a very weak signal of a few microvolts.

Figure 3 shows the block diagram and the complete architecture of this EFNMR spectrometer. The polarizing coil assembly includes two receiver coils (Rx) that are connected in an antiparallel configuration to cancel out environmental noise. The current flow through the polarizing coil is managed by a regulated power supply, controlled by a polarizing switch. The sample is placed inside one of the coils and the outputs of both Rx coils are fed into a differential amplifier through a relay-controlled switch.

Given the small magnitude of the resonance signal, on the order of microvolts, signal amplification is essential before further processing. Therefore, the signal is channeled through a fixed-gain ultra-low noise differential amplifier and a low noise amplifier, and then to a bandpass filter, with a cutoff frequency around the Larmor frequency, thus eliminating unwanted noise and frequencies.

The filtered signal undergoes an additional amplification stage before being acquired by the microcontroller ΣΔ-ADC. Finally, signal processing is implemented on these data to extrapolate and visualize the desired magnetic resonance information.

### 4.1. CAD Design and 3D Printing

Three-dimensional printing is a very useful tool for prototyping. It allows for the fabrication of parts designed using CAD programs. In this project, 3D printing has been used to manufacture the cylinders on which the coils are wound and the base that holds them, as well as other complementary parts that will be discussed below. To design the parts used in this project, the CAD program was used. This software is a comprehensive 3D design tool that combines parametric modeling, rendering, simulation, and computer-aided manufacturing functions in a single integrated package.

The first part designed was the cylinder of the polarizing coil. This cylinder has a height of 145 mm, an outer diameter of 90 mm, and an inner diameter of 80 mm. Two holes with a diameter of 2 mm were made 20 mm from the base to allow the coil wire to pass inside the cylinder. The second part designed was the cylinder of the receiving coils. This cylinder has a height of 140 mm, an outer diameter of 32 mm, and an inner diameter of 28 mm. Similarly to the polarizing coil cylinder, this cylinder also has two holes made 20 mm from the base, but this time with a diameter of 1.5 mm to allow for the coil wires to pass inside the cylinders. Figure 4 shows the CAD design of the polarizing coil and the receiving coil cylinders.

The three coils are aligned by means of a base that has grooves 15 mm deep and a width equivalent to the thickness of the walls of the cylinders where each of the coils is fitted, and this same base aligns the wires of connection of the coils to match the connections of the printed circuit.

In order to be able to modify the inclination of the coils, and to be able to align the terrestrial magnetic field in an optimal way, a support was designed, which holds the excitation coil and allows one to modify the angle with the base. The base of the equipment has a curved surface where the support can slide and two fixing screws, which allow us to secure the mounting in the desired position. The fixing screws used are plastic, so there is no need for any metal component that could alter the magnetic field near the coil. Figure 5 shows the CAD design of the various parts and the assembled model.

Once the design of all parts in CAD is completed, the files are transferred to a G-code translator. This kind of software is a sophisticated 3D model preparation software that enables the configuration and optimization of printing parameters. This process is essential before generating the G-code file required for 3D printing. The final result after the CAD design and printing is shown in Figure 6.

### 4.2. Polarizing Coil

The polarizing coil temporarily creates a polarizing magnetic field $B_{p}$. This field quickly turns off, leaving the magnetization instantly perpendicular to the Earth’s magnetic field around which precession begins [53,54]. The polarizing coil and its supporting structure must not contain any ferromagnetic material, see Figure 7. Naturally, the Earth’s field is very homogeneous, and if there is any ferromagnetic material near the coil, it will change the uniformity of the field, causing magnetic field inhomogeneities. To properly use the device during normal experiments, the entire system should be free of electromagnetic interference due to the weakness of the detected signal. However, since the goal of this project is to design a benchtop device, it has been designed to be robust under electromagnetic stress conditions. The purpose of the polarizing coil is to produce a magnetic field much greater than the Earth’s magnetic field. To polarize the sample, a current of approximately 10 A is passed through the coil for 10 s. Therefore, the wire for winding the polarizing coil must have a large-enough cross-section to avoid heating while carrying this current. The specifications chosen for the polarizing coil are given in Table 2.

The polarizing coil cylinder is wound on a commercial PVC (vinyl polychloride) tube. The polarizing coil is placed in the center of the coil holder, which is provided with holes for passing the connections to the PCB, placed under this holder. The base, inclination part, and coil support are made of PETG, avoiding the use of metal. The tilt platform should form an angle close to 50° relative to the base. The angle ensures that when $B_{p}$ is turned off, the magnetization is perpendicular to the Earth’s field immediately prior to signal detection. This geometrical arrangement maximizes the strength of the signal. The Earth field in southern Spain (University of Almeria) has an inclination of 50.57° and a declination of 0.59°.

### 4.3. Receiver Coils (Antenna)

The receiving coils ($R_{x}$) are responsible for collecting the signal from the polarized sample in precession. The signal is extremely low, requiring many iterations of the experiment to increase the signal-to-noise ratio (SNR). This process is known as signal averaging. The main problem affecting this device is noise, which is why the coils have been designed to minimize it. In addition to signal averaging, two nearly identical sensor coils are used to address this issue and maximize the SNR.

The two coils are wound in series but in opposite directions and placed on the same surface to cancel out noise (Figure 8). A real receiver coils built is shown in the Figure 9, this receiver coil is designed according its test tube. To receive the signal, the sample is placed in only one of the coils because if it is placed in both coils, the produced signals will cancel each other out, resulting in no signal being received at the output.

The most important requirement for correct winding is that both coils must be as identical as possible to perfectly cancel out environmental noise. This is called common-mode rejection and can be achieved if the number of turns is equal for both coils and the wire is not twisted at any point during winding. To verify the similarity of the coils, one method is to weigh the coils before and after winding. As with the polarizing coil, the material used for the cylinders on which the receiving coils are wound is a commercial PVC pipe.

As mentioned earlier, the device was designed to operate using the Earth’s magnetic field, so any nearby ferromagnetic material must be avoided. Metallic objects such as copper, aluminum, and brass should also be kept away from the coil to avoid induced eddy currents when $B_{p}$ is quickly turned off.

The wire’s cross-section is also an important parameter. With a thin wire, we have more turns and a stronger signal, but it also has higher resistance and, therefore, more Johnson noise $\propto \sqrt{R}$, which we do not want. On the other hand, with a thick wire, we have fewer turns and a weaker signal compared to a thin wire, but the noise is lower and the quality factor can be higher. The thermal noise follows the resistance relationship in volts:(11)$$V_{T}=\sqrt{4K_{B}TR\Delta f}$$
where $K_{B}$ is Boltzmann’s constant (J/K), *T* is the temperature in K, *R* is the resistance, and Δf is the bandwidth. The specifications of the receiving coils are summarized in Table 3.

To verify the proper functioning of the receiving coils, some preliminary tests were conducted. The coils were placed inside the polarizing coil, and a sinusoidal signal was applied to the polarizing coil using a function generator (Figure 10). This induces a sinusoidal wave in both receiving coils of the same amplitude and is approximately 180 degrees out of phase with the excitation source. Both waves cancel each other out, resulting in an almost null signal. All these signals were measured and analyzed in Matlab (Figure 11).

### 4.4. Power Circuit

This circuit is responsible for turning the power supply of the polarizing coil on and off with a signal from the microcontroller. This circuit had to be carefully designed because, for the experiment to work correctly, the coil needs to discharge as quickly as possible, and consequently, so does the magnetic field it generates. Therefore, no switch can be used to turn off the polarizing coil. Mechanical relays could be used to turn the polarizing coil on and off, but they are not the best option due to several disadvantages. The relay operates like a basic switch and can produce an electric arc when it is turned off, as the coil has high induced voltages. Relays also have a short lifespan and take a few milliseconds to switch. In our case, the polarizing coil needs to be turned on and off within microseconds. Therefore, the simplest solution to this problem is to use a power MOSFET with a shorter switching time.

The MOSFET used is the IPD090N03L. The maximum allowed collector-emitter voltage is 30 V, and the base-emitter voltage is 20 V. The maximum current it can drain is 40 A. The IPD090N03L has an extremely low internal on-resistance of 0.009 Ω. The positive power supply of +12 V is connected to one terminal of the polarizing coil, while the other terminal is connected to the MOSFET’s collector. The MOSFET emitter is connected to the ground. The MOSFET is operated by the microcontroller through a specific driver that boosts the voltage and current provided by the microcontroller to levels that ensure fast switching with the least amount of losses possible. Because high voltages are induced during the switching of the MOSFET, it is important to include a diode that absorbs these surges without damaging the MOSFET. Given the low resistance of the MOSFET, the amount of heat generated during switching can be dissipated by the PCB itself. The complete circuit schematic can be seen in Figure 12.

### 4.5. Receiver Circuit

The signal detected by the receiver coils is extremely small (on the order of microvolts). This signal needs to be converted by the microcontroller ADC to be processed by a computer, but it needs to be adapted; for this purpose, six different stages are designed as shown in Figure 13.

The first stage is a capacitor block. These capacitors act as tuners and are responsible for enhancing the signal before it reaches the instrumentation amplifier. This is achieved by placing these capacitors in parallel with the receiving coil. The value of the capacitor can be found using the resonant frequency formula, Formula (Equation 12).
(12)$$f=\frac{1}{2\pi \sqrt{LC}}$$

The frequency *f* is adjusted (adjusting *C*) to be as close as possible to the Larmor frequency of the proton, which is approximately 2 KHz. Here, *L* is the combined inductance of both coils, and its value is 5.12 mH. The optimal value of the tuning capacitor is 1.13 μF. This value is approximated by 1 μF and two 56 nF.

The second stage is a double relay controlled through the microcontroller that connects the output of the coils. The normal position of the relay switches is open. Each end of the receiving coils is connected to one of the relay switches, while the common point of the two coils is connected to the ground. When the microcontroller sends a signal to the transistor, it connects the relays to the +5 V power supply, causing the relay switches to activate, allowing the signal from the receiving coils to pass to the instrumentation amplifier. The circuit is shown in Figure 14.

Third, there is an ultra-low-noise instrumentation preamplifier. An INA848 is used, which is a high-precision instrumentation amplifier that requires no external components for a fixed gain of 2000. Since the signal is in microvolts, only low-noise amplifiers can be used. Both ends of the coils are connected to the input of the instrumentation amplifier, while the sample is placed in only one of the coils, as mentioned earlier. When the signal arrives, the signals from the two inputs of the instrumentation amplifier are subtracted, and the signal emitted by the proton is obtained.

The fourth stage is an amplifier stage, with a gain of 1000, built with one of the four operational amplifiers available in the OPA4189. In the fifth stage, signal filtering is performed, using the remaining three amplifiers to perform a band-pass filter of order 6, designed to attenuate the frequencies outside the range of interest for this application. Specifically, 1.8 kHz center frequency and a width of 1.5 kHz have been chosen. The filter response is shown in Figure 15.

The last stage transforms the signal to a differential mode and adapts the impedance values to those expected by the 16-bit ADC incorporated in the microcontroller. In this stage, the signal is shifted from referenced to GND to swing around $V_{ref}/2$, which is required to meet the input ranges of the ADC. Both the reference voltage and $V_{ref}/2$ are supplied by a precision voltage reference, in this case a REF2025.

The differential mode used significantly improves the conversion by increasing the noise immunity. No gain is introduced at this stage, so the total system gain is 2,000,000. The complete input schematic is shown in Figure 14.

### 4.6. Acquisition Sequence

The quality of the data obtained is strongly dependent on the precise timing of the different stages involved in the process. By design, this device alternates between the excitation stage, in which the polarizing coil is involved, and the relaxation stage, in which the receiver coils are involved. In this case, the acquisition process is fixed, and the timing parameters and the number of repetitions can be configured at the beginning of each experiment.

At the beginning of each scan of the experiment, the microcontroller activates the current conduction to the biasing coil. After the excitation time ($\tau_{1}$), the coil is turned off. Once a preset time ($\tau_{2}$) has elapsed, which is necessary for the field of the biasing coil to discharge, the relay closes and the receiver coils are connected to the amplifiers. Digitization begins after a second period ($\tau_{2}$), which ensures complete closure of the relay, the duration of which varies according to the number of samples required ($\tau_{3}$). At the end of the sampling, the relay opens and waits for the relaxation time ($\tau_{4}$). This process is repeated for the indicated number of times, improving the system’s ability to distinguish signal from noise.

Figure 16 details the temporal sequence that the device follows during each operation cycle. The time intervals associated with each phase of the sequence are detailed in Table 4.

### 4.7. Control and Communications

The control tasks of the equipment, the digitalization of the amplified and adapted signals in the previous section, and the communication with a computer are handled by an ARM microcontroller (STM32F373). This microcontroller has the required features for the correct operation of the unit, including a 16-bit-resolution analog–digital converter and USB communication for sending the large number of samples that need to be acquired. Of course, it has diverse input and output pins, timers, and SPI and I2C communication ports, among others. To enable USB communication, it is required to use a 72 MHz clock frequency obtained from an 8 MHz crystal and the internal PLL.

The required design is simple, having only the outputs corresponding to the activation of the polarizing coil and the relay, the analog inputs for the signal, and a communication port with the computer. The rest of the elements are auxiliary to achieve the correct operation of the instrument. Figure 17 shows the microcontroller section as well as the rest of the auxiliary elements of the circuit.

Since the coils need to be properly aligned with the terrestrial magnetic field, a 3-axis magnetometer is also incorporated, which is accessed by the microcontroller via the I2C bus. As the PCB is precisely aligned with the coils, it is convenient and practical to be able to use this magnetometer to adjust the position before starting the measurements, ensuring better results. The read values are visible from the program interface.

### 4.8. Computer Software

To ensure proper use of the unit, we developed basic control software in C#. This language for programming was chosen because it is driven by the extensive library support and the language’s ease of use, facilitating the enhancement and swift implementation of new experiments and functionalities. The main window (Figure 18) presents configurable experimental parameters, enabling the user to adjust the settings as required.

The number of samples to capture; according to this number and taking into account the fixed sampling frequency, the duration of this period is indicated, corresponding to $\tau_{3}$.Number of scans or number of times the acquisition cycle will be repeated. Corresponds to the parameter N.Recovery time, or time required after acquisition until a new cycle starts. Corresponds to $\tau_{4}$.Pulse time or polarizing coil activation time. Corresponding to $\tau_{1}$.


Figure 18The main window of the computer software is divided into two parts. On the left side, there is a control panel with textboxes and buttons for managing the acquisition process. On the right side, there is a primary plot window that displays either the detected signal or its Fast Fourier Transform (FFT) analysis.
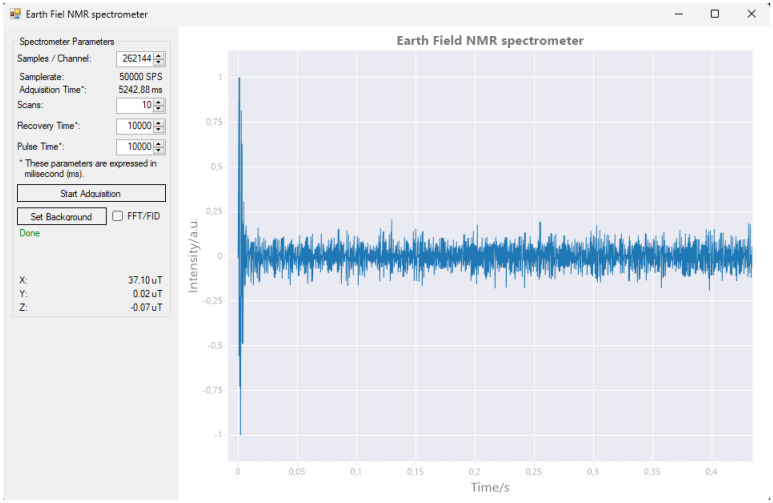



Although the software is intended to test the functionality of the machine, the ability to model the noise is incorporated, saving the results and allowing one to improve the behavior in subsequent experiments. This function is performed by running one or more scans without introducing any sample. The frequency response (FFT) obtained is stored in a file and used subsequently to suppress the background noise. The software also allows you to switch between the Fast Fourier Transform view (FFT) and the free-induction decay view.

### 4.9. Cost Analysis

The cost of the manufactured device is minimized due to its basic design and the use of readily available and inexpensive components. This approach ensures that the overall cost remains low while fulfilling the necessary functionality and performance requirements. The use of standard materials and simple manufacturing processes further enhances the cost effectiveness of the machine (see Table 5).

## 5. Results and Discussion

In this study, we successfully designed and built a complete assembly and electronic component EFNMR spectrometer from scratch. All parts, including the spectrometer framework, were manufactured with PETG material and 3D-printing technology with CAD design and 3D printing to ensure the device’s robust and accurate structure.

The electronic system includes a custom coil to generate the required magnetic field and a sensitive receiver coil to detect NMR signals. The STM32F373 microcontroller was used to control pulse sequences and read Analog–Digital Converter (ADC) signals from the receiver. This controller provides the computational power and flexibility necessary for accurate timing and signal processing. In Figure 19 is shown the successfully assembled device, integrating all components into a fully functional and cohesive EFNMR spectrometer.

The control software has a user-friendly interface and is written in C#. This C# application can be visualized in Figure 18 and Figure 20. This interface facilitates easy interaction with EFNMR spectrometers, allowing users to configure experimental parameters, begin measuring, and analyze the results. The software is designed to focus on simplicity and accessibility and enable users of different levels of expertise to use the spectrometer effectively.

All aspects of the project, including electronic designs, coil specifications, and control software, have been made available as open source resources on GitHub, https://github.com/fmarrabal/Earth-Field-NMR (accessed on 18 July 2024). This ensures that other researchers and enthusiasts can replicate, modify, and improve our work, fostering further advancements in the field of EFNMR spectroscopy.

The system was tested using standard samples, and the results demonstrated the capability of our EFNMR spectrometer to produce a ^1^H NMR spectrum. The results indicate that our EFNMR spectrometer can serve as a powerful tool for a wide range of applications, including chemical analysis, geophysical exploration, and educational purposes. Figure 20 shows a clear signal peak at approximately 1579 Hz, which is an expected result for Earth-field NMR. However, the baseline noise is quite prominent, suggesting that the overall SNR is low. The noise appears to be relatively high compared to the signal peak. In an ideal NMR experiment, the signal should stand out distinctly above the noise floor. Here, the noise level is only marginally lower than the intensity of the peak, which implies that the SNR could be improved. In Earth-field NMR, the sensitivity is inherently lower than in high-field NMR because of the significantly smaller magnetic field strength. This leads to a reduced net magnetization, which directly impacts the strength of the detected signal. To improve the sensitivity, it is possible to perform the following:Increase the number of scans: averaging more scans can improve the SNR by reducing random noise.Optimize pulse parameters: adjusting pulse times and recovery times to ensure the optimal alignment of the spins prior to detection can enhance the signal.Signal averaging and background subtraction: techniques such as averaging multiple acquisitions and subtracting background noise can also be helpful in enhancing the detectable signal.Magnetization enhancement techniques: implementing techniques such as hyperpolarization or using stronger polarizing coils might also improve initial magnetization, thus increasing the signal intensity.

This device and software are designed to optimize sensitivity through a variety of advanced techniques. They are capable of setting the number of scans to improve the signal-to-noise ratio (SNR) by averaging multiple acquisitions, thus reducing random noise. The system can also fine-tune pulse times and recovery periods to ensure optimal spin alignment before detection, which enhances the detected signal’s strength. In addition, the software facilitates signal processing methods, such as background subtraction and real-time signal averaging, to further enhance the clarity of the detected peaks. Although these capabilities significantly improve the sensitivity of the NMR measurements, it is important to note that the system does not incorporate magnetization enhancement techniques, which would require external interventions beyond the microcontroller and software scope.

In summary, the development of this EFNMR spectrometer demonstrates the feasibility of constructing high-quality, low-cost NMR systems using readily available materials and open-source technology. Our work lays the foundation for future innovations in portable and accessible NMR spectroscopy oriented to AI-enhanced low-field NMR technology.

## Figures and Tables

**Figure 1 sensors-24-05537-f001:**
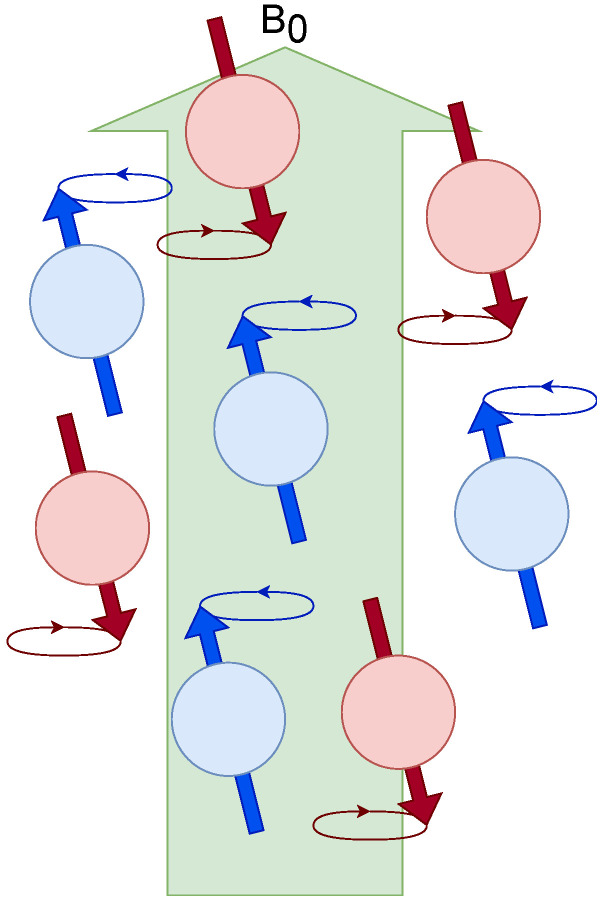
Energy levels considered before an external magnetic field $B_{0}$. Populations are generated that are quantized with two energy levels.

**Figure 2 sensors-24-05537-f002:**
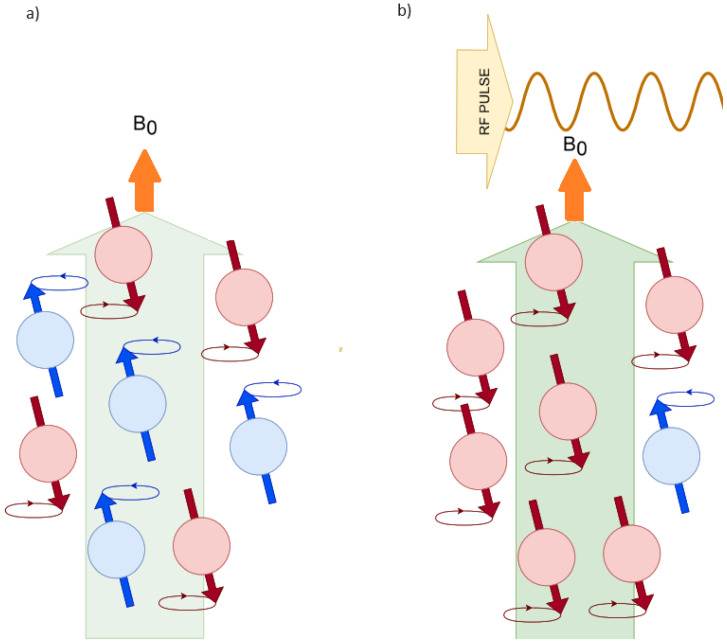
(**a**) Spin population distribution before the application of an RF pulse or polarizing coil in an external magnetic field Bo. Not all spins are aligned with Bo, leading to net magnetization. (**b**) Spin population distribution after the application of the RF pulse or polarizing coil. The pulse causes some spins to transition to a higher energy state, partially inverting the spin population, but not all spins are fully inverted. This diagram reflects the Zeeman energy levels and the alignment of nuclear spins with the external magnetic field. A radiofrequency (RF) pulse perturbs the nuclear spins from their equilibrium state, which are “on resonance”.

**Figure 3 sensors-24-05537-f003:**
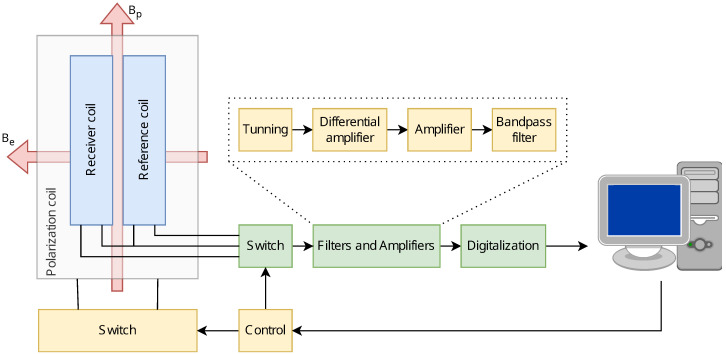
Block diagram of the electrotechnical device for nuclear spin excitation in the Earth’s magnetic field. Bp and Be represent the polarization magnetic field and the Earth’s magnetic field, respectively.

**Figure 4 sensors-24-05537-f004:**
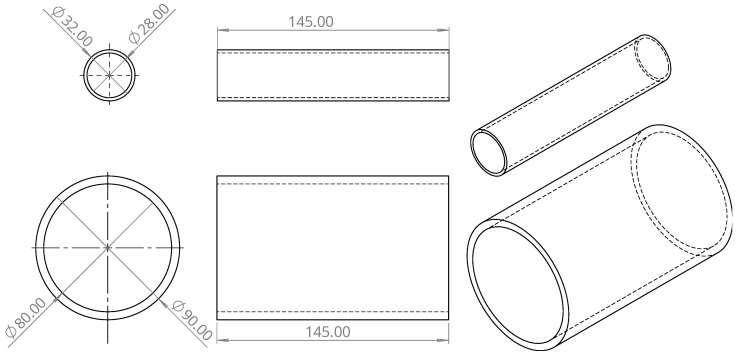
CAD design of the polarizing and receiving coil cylinders.

**Figure 5 sensors-24-05537-f005:**
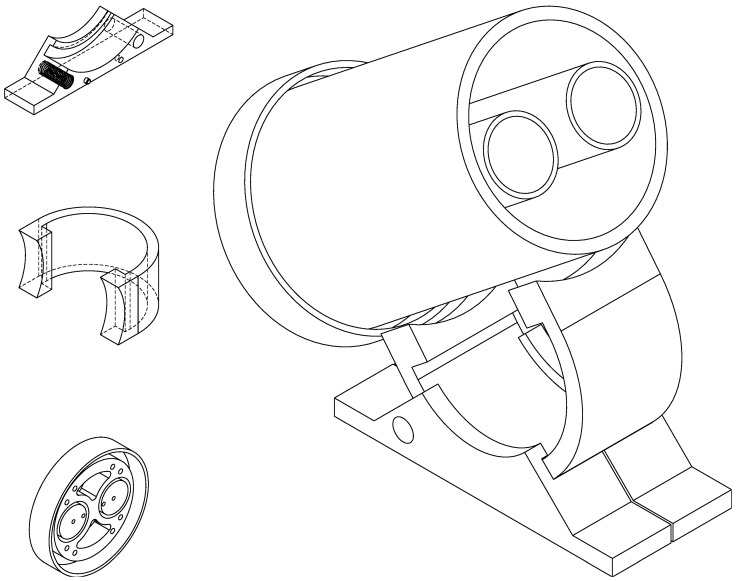
CAD design of the bases that support the coils and assembled model.

**Figure 6 sensors-24-05537-f006:**
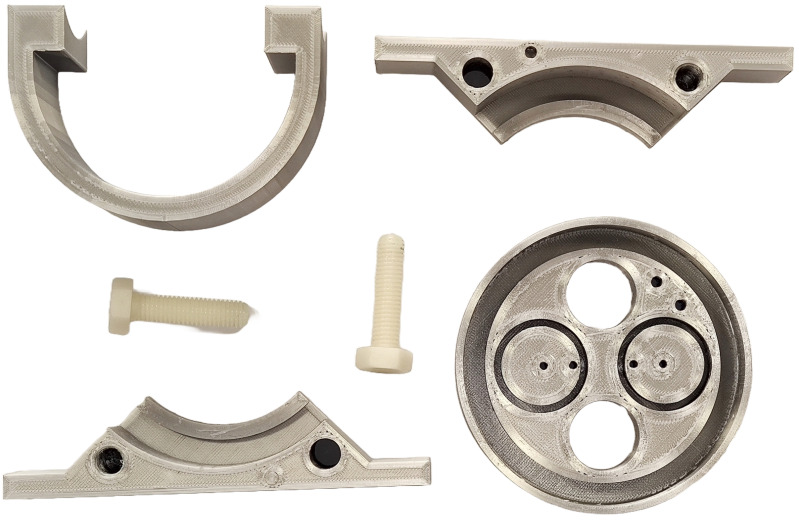
The 3D-printed parts with PETG and Nylon M10 threads.

**Figure 7 sensors-24-05537-f007:**
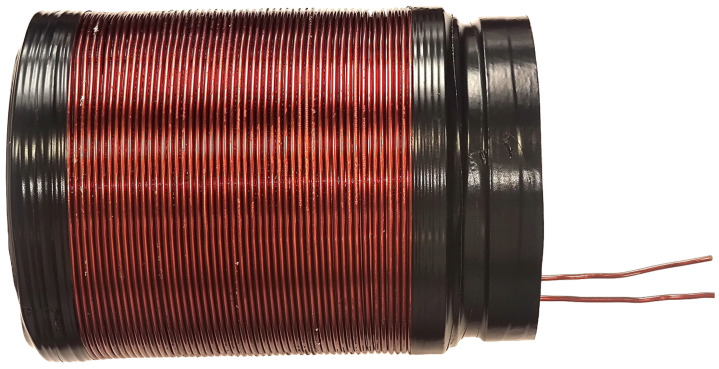
Wounded polarizing coil.

**Figure 8 sensors-24-05537-f008:**
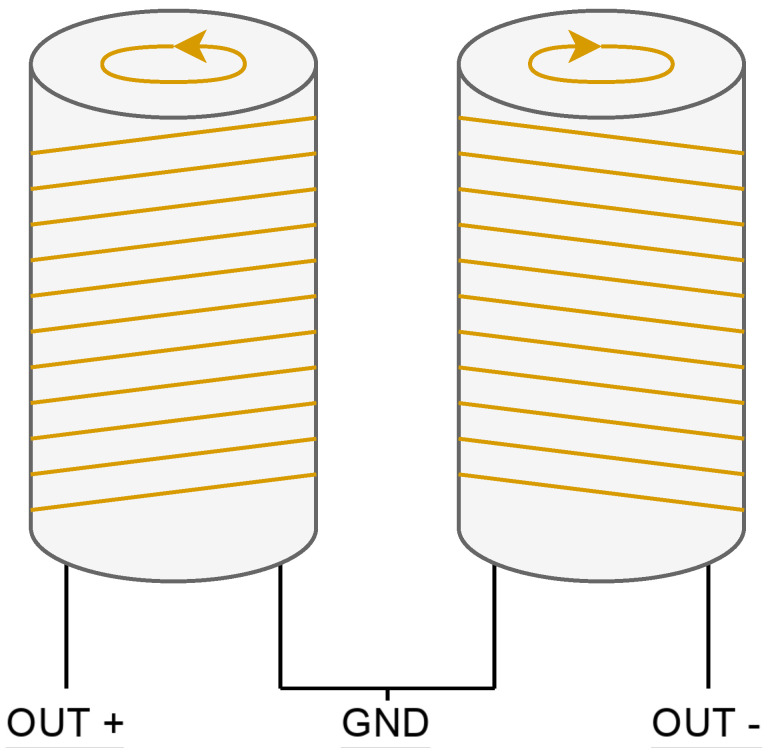
Winding diagram of the receiver coils.

**Figure 9 sensors-24-05537-f009:**
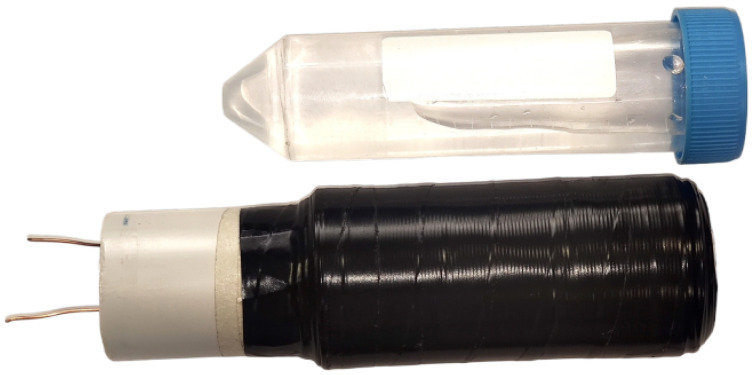
Wounded receiver coil and test tube.

**Figure 10 sensors-24-05537-f010:**
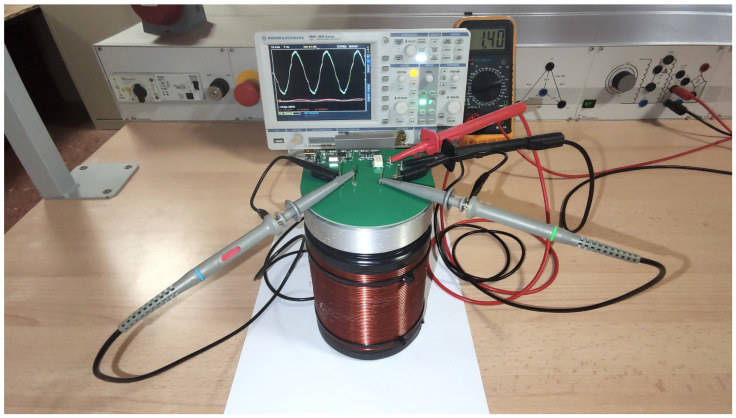
Testing of receiver coils in the electrotechnical laboratory.

**Figure 11 sensors-24-05537-f011:**
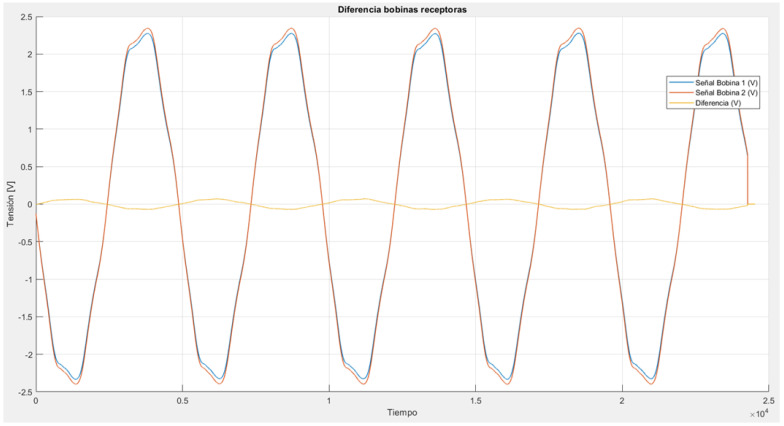
Test results give a good response of the avoiding signal between both coils. Only a little 180° dephased signal was measured, which ensures the cancelation of the signals.

**Figure 12 sensors-24-05537-f012:**
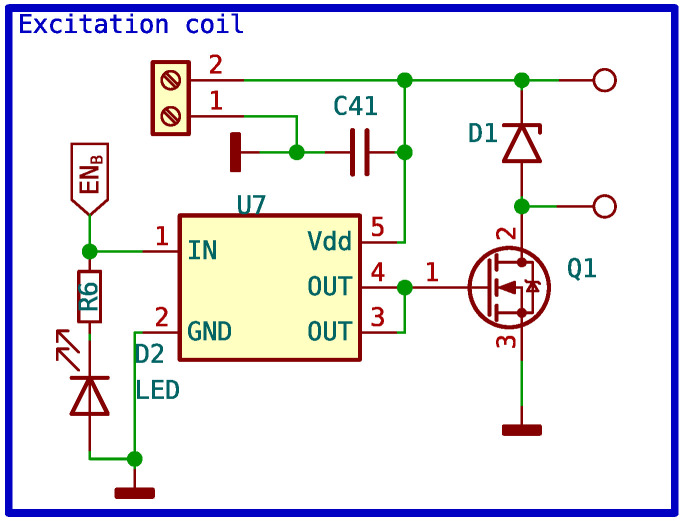
Circuit schematic of excitation coil control.

**Figure 13 sensors-24-05537-f013:**
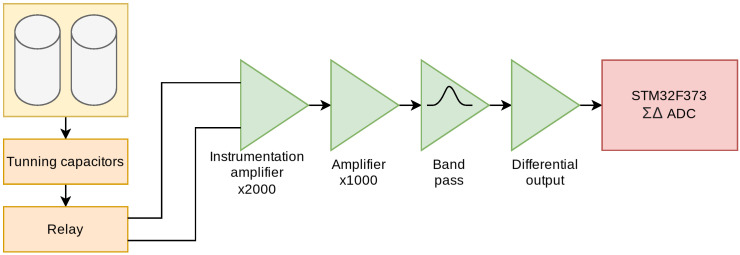
Block diagram of the amplification and filtering circuit.

**Figure 14 sensors-24-05537-f014:**
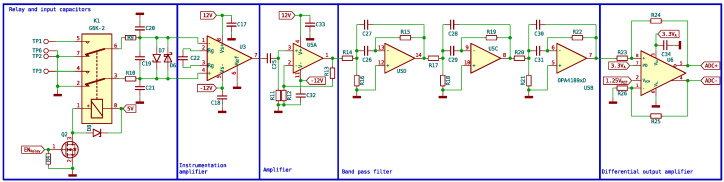
Switching amplification and filtering input stages circuit.

**Figure 15 sensors-24-05537-f015:**
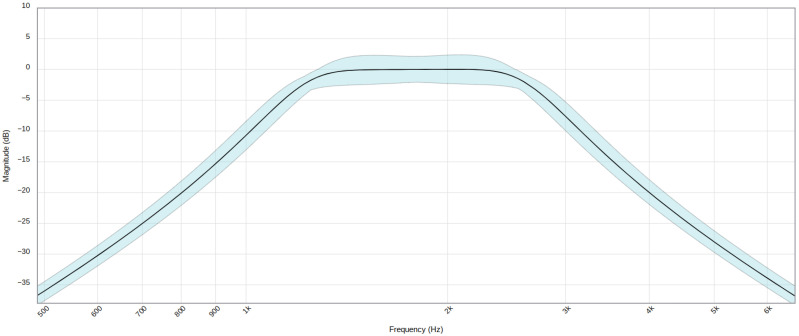
Filter stage frequency response. Blue band shadow includes components’ tolerances.

**Figure 16 sensors-24-05537-f016:**
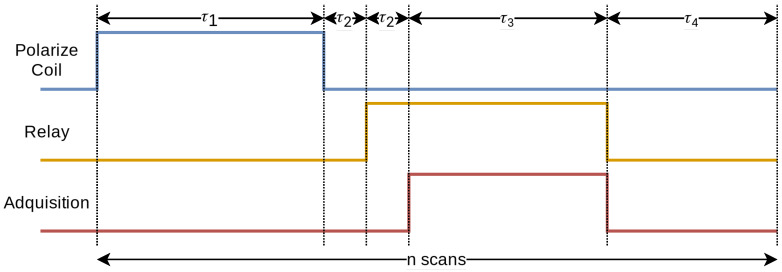
Sequence of pulses followed by the device as a function of time.

**Figure 17 sensors-24-05537-f017:**
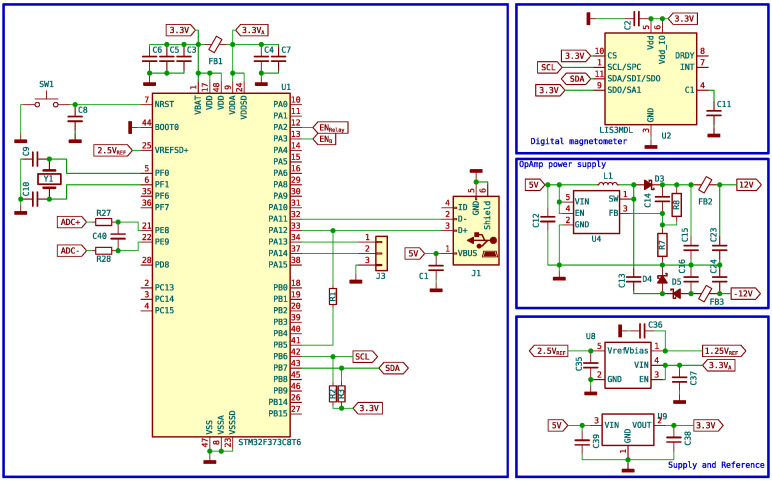
Microcontroller schematic and auxiliary components.

**Figure 19 sensors-24-05537-f019:**
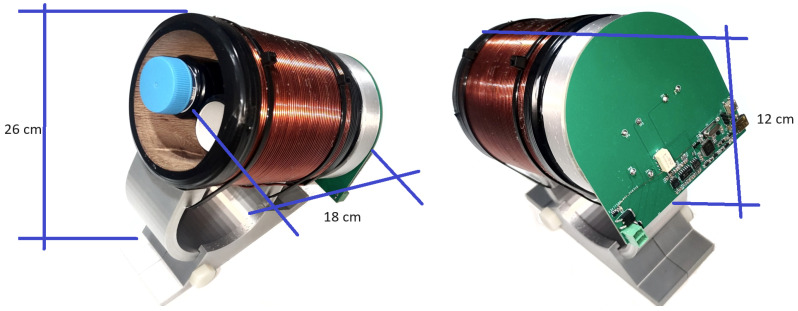
The final device was successfully assembled, integrating all components into a cohesive and fully functional EFNMR spectrometer.

**Figure 20 sensors-24-05537-f020:**
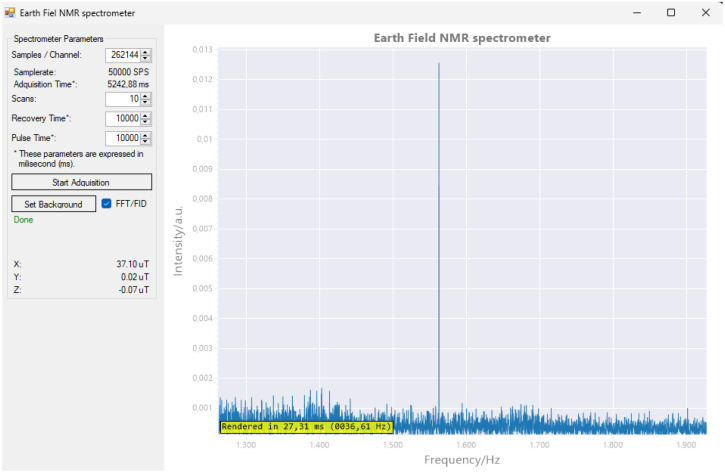
^1^H MR spectrum test was successfully conducted. The magnetometer detected a magnetic field strength of 37.10 μT, resulting in a resonance frequency of 1579 Hz for the ^1^H nuclei.

**Table 1 sensors-24-05537-t001:** Model of equilibrium and non-equilibrium in the resonant process of nuclear spin.

Equilibrium	Non-Equilibrium
min (ΔE−Δϕ)	max (ΔE−Δϕ)
$M_{z}=M_{0}$	$M_{z}=M_{0}\cos\left(\phi_{p}\right)$
$M_{x}=0$	$M_{x}=0$
$M_{y}=0$	$M_{y}=M_{0}\sin\left(\phi_{p}\right)$

**Table 2 sensors-24-05537-t002:** Specifications of the polarizing coil.

Coil	Specification
Enameled copper wire	1.7 mm or 14 AWG
Inner diameter of the coil	80 mm
Outer diameter of the coil	90 mm
Height of the coil	145 mm
Number of wire layers	3
Number of turns per layer	60
Approximate inductance	3.4 mH
Approximate resistance	0.8 Ω

**Table 3 sensors-24-05537-t003:** Specifications of the receiver coil.

Coil	Specification
Enameled copper wire	0.812 mm or 21 AWG
Inner diameter of the coil	25 mm
Outer diameter of the coil	40 mm
Height of the coil	100 mm
Number of wire layers	4
Number of turns per layer	115
Approximate inductance per coil	2.56 mH
Approximate resistance per coil	2.5 Ω

**Table 4 sensors-24-05537-t004:** Times for each stage of the sequence.

Stage	Symbol	Minimum	Maximum	Default
Polarizing coil activated	$\tau_{1}$	1 s	32 s	10 s
Pre-acquisition	$\tau_{2}$	-	-	100 ms
Acquisition	$\tau_{3}$	1000 samples	5,000,000 samples	250,000 samples
Acquisition sample rate	-	-	-	50,000 SPS
Waiting between experiments	$\tau_{4}$	1 s	32 s	5 s
Number of scans	N	1	5000	650

**Table 5 sensors-24-05537-t005:** Cost breakdown of EFRMN device components.

Item	Quantity	Cost
PETG Filament	1 kg	24 €
PCB Fabrication	-	10 €
Components	-	100 €
Copper Wire 1.7 MM	70 m	30 €
Copper Wire 0.8 MM	120 m	25 €
Wiring	-	10 €
Power supply	1	30 €
	Total	229 €

## Data Availability

The repository https://github.com/fmarrabal/Earth-Field-NMR (accessed on 18 July 2024) provides complete device schematics, PCB design, microcontroller firmware, and computer software for the device.

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
