# Peer review of "Development of an Earth-Field Nuclear Magnetic Resonance Spectrometer: Paving the Way for AI-Enhanced Low-Field Nuclear Magnetic Resonance Technology"

_sensors, 2024, doi:10.3390/s24175537_

Round 1

Reviewer 1 Report

Comments and Suggestions for Authors

This is a very interesting project elaborating on earth field NMR using polarization in lieu of the 90 degree pulse of high field spectrometers. This is also very useful in terms of the github repository and this now is making this setup accessible for use in the field.

The paper is also well written and provides the base level required introduction to NMR and this will be useful to those wanting to implement this who have not come through for example traditional organic chemistry routes to NMR. As such I believe it fulfils a more general role.

The study itself is multifaceted, from the engineering to the precision electronics through to software and of course the chemistry involved. My feeling is that this strengthens this considerably.

My issues with this however are as follows.

1. In terms of analysis, 1H signals normally are in the range 0-10ppm. In figure 20 this would be a variation of 0.015Hz - which I do not think will be visible. Please elaborate on the usefulness of the technique in this context?

2. Are you certain that the signal is from the sample (e.g. when sample is absent, is the signal still present)? Are you sure that this is not from transients in the circuit? What would really strengthen this for me is the NMR on this instrument of other nuclei, specifically 31P and 19F, which would show a signal at 639Hz and 1486Hz in Figure 20.

Author Response

Thank you for your thoughtful and encouraging feedback on the project. I’m glad that you find the work interesting and recognize the potential of making Earth field NMR more accessible through the provided setup and GitHub repository. I appreciate your acknowledgment of the multifaceted nature of the study and its broader educational value.

Comments 1:

In conventional high-field NMR spectroscopy, the chemical shift range for 1H nuclei (protons) is typically 0-10 ppm, corresponding to a frequency range of several hundred to several thousand Hz, depending on the magnetic field strength of the spectrometer. In an NMR system based on Earth's magnetic field (~50 µT. theorecally), in our case arround 37µT , the resonance frequency for protons is equal or less to 2 kHz. Therefore, the chemical shift range of 0-10 ppm would correspond to a very narrow frequency range, approximately 0.015 Hz as you mentioned.

Expressing the chemical shift in terms of ppm in this context does not make practical sense due to the extremely low resolution of the frequency in an Earth's magnetic field NMR system. To observe and resolve a 1 peak in this NMR spectrometer without differeces between each other, we need to consider a much broader chemical shift range, possibly in the order of thousands of ppm. That's why we use frequency scale and not he ppm scale. This significantly increase the frequency difference and allow the peak to be detected within the limitations of the Earth's magnetic field. This adjustment acknowledges the low resolution inherent to the system and aligns the expectations with what the instrument can realistically achieve.

Comments 2:

As you correctly pointed out, verifying that the signal originates from the sample is very important.  In response to this, we conducted control experiments with the sample removed. These tests confirmed that the signal does not persist when the sample is absent, strongly indicating that the detected signals are indeed due to nuclear resonance from the sample rather than transients or noise in the circuit.

Regarding your suggestion to test with other nuclei such as 31P and 19F, it's worth noting that 19F and 1H have very similar gyromagnetic constants, which makes 19F particularly suitable for this type of experiment. Moreover, that depends on its natural abundance as well. We fully intend to conduct these experiments, but they are planned for the next phase of the project using a very low magnetic field NMR spectrometer, which will operate at 2 MHz. This upcoming spectrometer will offer higher sensitivity and better resolution, allowing us to perform these critical tests with 19F and other nuclei such as 31P, further enhancing the robustness and versatility of our NMR studies. Our goal is AI-driven very low magnetic field NMR spectrometer gives a perfomance as a High Resolution NMR spectrometer.

Thank you again for your valuable feedback. Your insights have been very helpful in refining the scope and focus of this work, and I look forward to incorporating these suggestions into the next phase of the project.

Reviewer 2 Report

Comments and Suggestions for Authors The review report on paper entitled ‘Development of an Earth Field NMR Spectrometer: Paving theWay for AI-Enhanced Low Field NMR Technology' by Eduardo Viciana, Juan Antonio Martinez Lao, Emilio Lopez-Lao, , Ignacio Fernandez , and Francisco Manuel Arrabal-Campos        In the paper the authors consider implications of the Nuclear Magnetic Resonance to the Earth’s magnetic field. The detailed and extended overview of NMR and applications is given in the manuscript. In their work the authors concentrate on proton NMR, i.e., the 1H isotope. The paper materials are more methodological, give practical hints for an enhancement of NMR technique sensitivity for small fields, and warrant to express publication in J. 'Sensors' .

Author Response

Thank you very much for your positive feedback on our manuscript. We appreciate your recognition of the methodological focus of our work and your acknowledgment of its practical contributions to enhancing NMR sensitivity in low-field environments.

We are particularly pleased that you found the detailed overview of NMR and its applications to be comprehensive and valuable. Our goal was to provide a solid foundation for those interested in exploring Earth field NMR, particularly with an emphasis on practical enhancements for sensitivity, and we are glad that this aspect of our work resonated with you.

Your support for our paper's publication in Sensors is highly appreciated. We believe that the insights and methodologies presented in our research will contribute meaningfully to the field of low-field NMR technology and its future advancements, especially with the integration of AI.

Thank you once again for your encouraging review, and we look forward to the opportunity to share our work with the broader scientific community through this publication.

Reviewer 3 Report

Comments and Suggestions for Authors

This is good paper presenting the design of an NMR earth field spectrometer from scratch. I highly recommended it for publication provided the following points are addressed :

General comments :

The country should be specified in the address...

The introduction is very complete and provide a detailed history of NMR, but maybe all of these details (4 pages) are not fully relevant for an article describing a new NMR spectrometer. I would suggest focusing more on the NMR instrument and earth field NMR history (which is only half a page). For example is the design with two receiver coil very popular for this kind of instrument? Is this two coils design necessary ?

The cost should be mentioned and discussed.

Introduce the term FID in the description of the NMR experiment (for example around line 350). Show a graphical representation (eq 3/4/5).

The use of the polarizing coils does not correspond to the experiment describe in section 3 (principle). In the experiment performed  there is no RF pulse. One could think of using the receiving coil also for excitation why is the polarizing coil method preferred ? This should be explained.

What is the sample for the measured spectrum ? What is the resolution on the spectrum (figure 20). Discussed signal to noise and sensitivity issue.

Figure 19 : it would be good to have an idea of the actual size on the picture.

Figure 2 is not clear after pulse not all spin are inverted, nor during pulse. Add a) and b) to distinguish between the two situation. in Caption : Provide title and clarify :  Zeeman energy levels and determine

Typos :

Line 236 : 19 should be in superscript

Figure 2 is not clear after pulse not all spin are inverted, nor during pulse. Maybe this figure should be replace by one showing the use of a polarizing coil ? Add a) and b) to distinguish between the two situation. in Caption : Provide title and clarify :  Zeeman energy levels and determine

Typos :

Table 4 Samplerate-> Sample rate.

Line 279 : Conventional spectrometer is repeated twice, I think the authors means earth field instead conventional for the first spectrometer.

Line 575 Acquisition not adquisition

Comments on the Quality of English Language

I am not an English native speaker myself, but I think the English could be slightly improved.

Author Response

Thank you for your constructive feedback on our manuscript. We appreciate your suggestion to streamline the introduction and focus more on the aspects directly relevant to the Earth field NMR spectrometer. In response to your comments:

  • We have added the country.
  • We have reduced the NMR history section to one page, as you suggested.
  • We have added the point 4.9 with the cost breakdown of EFRMN device components.
  • As the comments suggest, we have introduced the FID concept when we explain the relaxation process  in the paragraph before of equations 3/4/5. I have added some references where the graphical representation of these equations have well plots.
  • We have changed the figure 19 adding dimensions of the device.
  • You are correct in noting that the experiment described in Section 3 does not involve an RF pulse in the traditional sense. Instead, we utilize a polarizing coil to achieve the initial magnetization, which serves as an alternative to the conventional RF excitation pulse used in high-field NMR. Clarification of the Polarizing Coil Method:

    • Enhanced Magnetization: In the context of Earth field NMR, where the magnetic field strength is extremely low, the thermal polarization of the spins is also very weak. The polarizing coil provides a stronger, more uniform magnetic field during the polarization phase, which effectively increases the initial magnetization of the sample before the signal detection phase. This increased magnetization is crucial for achieving a detectable signal in low-field conditions.

    • Decoupling of Functions: Using separate coils for polarization and reception allows us to optimize each function independently. The polarizing coil can be specifically designed to generate a strong, uniform magnetic field, while the receiving coil can be optimized for maximum sensitivity and signal-to-noise ratio during the detection phase. This separation of tasks enhances the overall performance of the system.

    • Minimizing Artifacts: If the same coil were used for both excitation and reception, there could be a risk of introducing artifacts or residual signals from the excitation phase into the detection phase. The use of a dedicated polarizing coil helps to mitigate this risk, leading to cleaner and more accurate signal acquisition.

  • This Polarizing Coil Method is explained at the end of the point 3 adding a paragraph. 
  • The figure 20 is an experiment of a water sample.  The figure shows a clear signal peak at approximately 1579 Hz, which is an expected result for Earth field NMR. However, the baseline noise is quite prominent, suggesting that the overall SNR is low. The noise appears to be relatively high compared to the signal peak. In an ideal NMR experiment, the signal should stand out distinctly above the noise floor. Here, the noise level is only marginally lower than the intensity of the peak, which implies that the SNR could be improved.  In Earth field NMR, the sensitivity is inherently lower than in high-field NMR because of the significantly smaller magnetic field strength. This leads to a reduced net magnetization, which directly impacts the strength of the detected signal. I have add some comments in the point 5. The figure 20 has a FFT resolution of 0.1907 Hz.
  • All typos were corrected.
  • The figure 2 was changed and divided in two figures (a) and (b).

Thank you again for your valuable input, which has been instrumental in refining the focus and clarity of our manuscript. We are confident that these revisions will strengthen the overall impact of the paper.